# Antiproliferative Fatty Acids Isolated from the Polypore Fungus *Onnia tomentosa*

**DOI:** 10.3390/jof8111163

**Published:** 2022-11-03

**Authors:** Hooi Xian Lee, Wai Ming Li, Jatinder Khatra, Zhicheng Xia, Oleg Sannikov, Yun Ling, Haoxuan Zhu, Chow H. Lee

**Affiliations:** 1Department of Chemistry and Biochemistry, Faculty of Science and Engineering, University of Northern British Columbia, Prince George, BC V2N 4Z9, Canada; 2Department of Chemistry, University of British Columbia, Vancouver, BC V6T 1Z1, Canada

**Keywords:** *Onnia tomentosa*, *Hymenochaetaceae*, linoleic acid, oleic acid, fatty acids, antiproliferative, autoxidation

## Abstract

*Onnia tomentosa* is a widespread root rot pathogen frequently found in coniferous forests in North America. In this study, the potential medicinal properties of this wild polypore mushroom collected from north–central British Columbia, Canada, were investigated. The ethanol extract from *O. tomentosa* was found to exhibit strong antiproliferative activity. Liquid–liquid extraction and bioactivity-guided fractionation, together with HPLC-MS/MS and 1D/2D NMR analyses of the ethanol extract of *O. tomentosa*, led to the identification of eight known linoleic oxygenated fatty acids (**1.1**–**1.4** and **2**–**5**), together with linoleic (**6**) and oleic acids (**7**). The autoxidation of linoleic acid upon isolation from a natural source and compound **5** as an autoxidation product of linoleic acid are reported here for the first time. GC-FID analysis of *O. tomentosa*, *Fomitopsis officinalis*, *Echinodontium tinctorium*, and *Albatrellus flettii* revealed linoleic, oleic, palmitic, and stearic acids as the major fatty acids. This study further showed that fatty acids were the major antiproliferative constituents in the ethanol extract from *O. tomentosa*. Linoleic acid and oleic acid had IC_50_ values of 50.3 and 90.4 µM against human cervical cancer cells (HeLa), respectively. The results from this study have implications regarding the future exploration of *O. tomentosa* as a possible edible and/or medicinal mushroom. It is also recommended that necessary caution be taken when isolating unstable fatty acids from natural sources and in interpreting the results.

## 1. Introduction

Mushrooms, the fruiting bodies of fungi, have been known for a long time in many different cultures around the world to possess medicinal properties. However, unlike in Asia and parts of Eastern Europe, relatively few studies have been conducted exploring mushrooms native to North America for potential medicinal properties. For instance, as of 2020, only 79 species of mushrooms native to North America had been investigated for their medicinal properties, and out of these, 48 species or 60% have been found to possess bioactivities that have not been previously reported [1]. Seventeen new bioactive compounds, comprising 10 small molecules, six polysaccharides, and one protein, had already been isolated from 16 selected species [1]. Such information supports the notion that mushrooms remain largely an untapped resource for drug discovery. To this end, we have continued our efforts to explore mushrooms native to British Columbia, Canada, for their potential medicinal properties and to isolate their respective bioactive compounds [2,3,4,5,6].

The objective of this study was to explore a polypore fungus called *Onnia tomentosa* (Fr.) P. Karst, formerly known as *Inonotus tomentosa* (Fr.) Teng. Many polypores are considered non-toxic and some are well-known for their medicinal properties, with notable examples such as *Ganoderma lucidum* and *Trametes versicolor* [7]. *O. tomentosa* is considered one of the most widespread root rot pathogens in boreal and sub-boreal forests in North America, and its patterns of colonization have been extensively studied [8]. To date, studies conducted on *O. tomentosa* have mainly focused on the root diseases caused by this fungus [9] and on the site and soil characteristics related to the incidence and spread of the fungus [10]. Prior to this study, *O. tomentosa* had never been investigated for its bioactivity; therefore, there is no information on its chemical composition.

Given the lack of study as described above, it was of interest to investigate whether *O. tomentosa* contains compound(s) with potential anticancer properties. Therefore, the polypore was collected from north–central BC and its extracts were screened for bioactivities related to cancer treatment, including antiproliferative and immunomodulatory activities. In this study, the ethanol extract of *O. tomentosa* showed strong antiproliferative activity. Liquid–liquid extraction, size exclusion chromatography, and HPLC-MS, followed by HPLC-MS/MS and NMR analyses, led to the conclusion that fatty acids, notably linoleic and oleic acids, were the major antiproliferative compounds in *O. tomentosa*. This study shows for the first time the autoxidation of linoleic acid isolated from a natural source. This has important implications for future studies on isolating bioactive fatty acids from natural sources. 

## 2. Materials and Methods

### 2.1. Material and Chemicals

Human cervical cancer cell (HeLa) was purchased from the American Type Culture Collection. Fetal bovine serum was obtained from Life Technologies Inc. (Waltham, MA, USA) and Minimal Essential Medium was obtained from LONZA (Walkersville, MD, USA). The Sephadex LH-20 was obtained from GE Healthcare (Uppsala, Sweden). Linoleic acid (18:2, >98% pure), oleic acid (18:1, 99% pure), 3-(4,5-dimethylthaizaol-2-yl)-2,5-diphenyl tetrazolium bromide (MTT), deuterium oxide (D_2_O), doxorubicin, and lipopolysaccharide were purchased from Sigma (St. Louis, MO, USA). HPLC-grade acetonitrile and methanol, as well as ACS-grade ethanol and hexane, were obtained from BDH (Mississauga, ON, Canada). ACS-grade dichloromethane and diethyl ether were obtained from Macron Fine Chemicals (Avantor, PA, USA). ACS-grade chloroform and LCMS-grade formic acid were obtained from Fisher Scientific (Ottawa, ON, Canada).

### 2.2. Identification of Mushroom Species

All four mushrooms used in this study, *Fomitopsis officinalis* (CL115), *Echinodontium tinctorium* (CL160), *Albatrellus flettii* (CL206), and *Onnia tomentosa* (CL83 and CL312), were collected from northern BC, Canada. The details of the collection sites and years are listed in Appendix A. The mushroom collections were initially identified based on morphological characteristics. Voucher specimens for these collections were deposited at the University of Northern British Columbia, Canada. All collections were confirmed via DNA sequencing. Genomic DNA extraction and polymerase chain reaction (PCR) amplification were performed as described previously [11,12]. The DNA sequences of the Internal Transcribed Spacer 2 (ITS2) region were aligned and edited using CLC Main Workbench (Qiagen, Carlsbad, CA, USA) and then entered into the Basic Local Alignment Search Tool (BLAST) to determine the closest sequence identity in GenBank (Appendix A).

### 2.3. Preparation of Crude Extracts from Onnia tomentosa

The fungal specimens were dried and then crushed into powder form and sequentially extracted into four extracts as previously described [11,12]. For large-scale extraction, powdered *O. tomentosa* CL312 (87 g) was first extracted with 80% ethanol for 3 h at 65 °C. The solution, filtered through Whatman paper No. 3, was then referred to as E1, whereas the residue was subjected to the second step: 50% methanol extraction for 3 h at 65 °C. This filtrate, designated as E2, was retained while the residue was subjected to the third step: water extraction for 6 h at 65 °C. The filtrate from the water extraction step was referred to as E3. Finally, this residue was subjected to the final step: 5% sodium hydroxide (NaOH) extraction at 65 °C for 6 h. The filtered solution from this step was referred to as E4. Both E1 and E2 liquid extracts were concentrated using a rotary evaporator before being subjected to lyophilization to yield solid extracts of 13.2 g (15.2%) and 5.3 g (6.1%), respectively. All lyophilized extracts were reconstituted in water at 20 mg/mL and filter-sterilized using a 0.2 μm filter (Sarstedt, Quebec), before they were assessed for antiproliferative and immunomodulatory activities as described below. 

### 2.4. Fractionation of Extract from Onnia tomentosa

E1 extract from *O. tomentosa* was chosen for further studies because it possessed potent antiproliferative activity. E1 was first subjected to liquid–liquid extraction. The E1 extract (2.1 g) was resuspended in water and partitioned with hexane (HEX) to obtain hexane-soluble material. The water fraction was further partitioned with diethyl ether (DEE) and chloroform (CHCl_3_). Each solvent layer was evaporated to yield solid extracts-H_2_O (1.7 g, 77%), HEX (0.15 g, 6.9%), DEE (0.12 g, 5.5%) and CHCl_3_ (0.06 g, 2.9%). All the layers except the H_2_O layer showed antiproliferative activity. 

HEX and DEE layers were subjected to Sephadex LH-20 column chromatography (70 cm × 16 mm i.d., in methanol) to yield 36 fractions each (5 mL per fraction). Fractions 10–20 exhibited antiproliferative activity and were subjected to reversed-phase column purification with an analytical HPLC Agilent Infinity Lab Poroshell 120 EC-C18 (4.6 mm × 100 mm × 2.7 µm) with a gradient mobile phase composed of an H_2_O solution of 0.1% formic acid (solvent A) and CH_3_CN containing 0.1% formic acid (solvent B) at a flow rate of 1 mL/min and wavelength (λ) at 230–280 nm. The gradient elution program was set as follows: 0 min (55% B), 17 min (46.5% B), 21 min (70% B), 23 min (100% B), and 35 min (100% B). Four semi-purified/purified compounds were obtained and later named mixture **1** (retention time, RT 10.4 and 10.7 min), compound **3** (RT 15.6 min), **4** (RT 17.0 min), and **5** (RT 20.3 min). Semi-purified mixture **1**, which exhibited of two major peaks, was later subjected to a second purification step using the same column but with slight modifications in the solvent system as follows: 0 min (55% B), 15 min (47.5% B), and 17 min (100% B). Using the same active fraction, two compounds, **6** (RT 23.1 min) and **7** (RT 28.4 min), were detected at 200 nm and later subjected to a new chromatographic separation method using a Phenomenex Phenyl-hexyl column (4.6 mm × 250 mm × 5µm) with a gradient mobile phase composed of H_2_O solution of 0.1% formic acid (solvent A) and CH_3_CN containing 0.1% formic acid (solvent B) at a flow rate of 1 mL/min. The mobile phase gradient was set as follows: 0 min (55% B), 15 min (70% B), 31 min (70% B), 32 min (100% B), and 35 min (100% B). Isolated compounds were analyzed via NMR and HPLC-MS/MS analysis as described below.

10-hydroxy-(8*E*)-octadecenoic acid (**5**). White powder; C_18_H_34_O_3_; ^1^H NMR (600 MHz, D_2_O) δ 0.86 (t, 3H, H-18), 1.21–1.61 (m, 20H, H-4 to H-6, H-11 to H-17), 1.45–1.61 (m, 2H, H-3), 2.04 (q, *J* = 7.3 Hz, 2H, H-7), 2.17 (t, *J* = 7.5 Hz, 2H, H-2), 4.08 (q, *J* = 7.0 Hz, 1H, H-10), 5.46 (dd, *J* = 15.4, 7.5 Hz, 1H, H-9), 5.70 (dt, *J* = 15.5, 6.9, 6.9 Hz, 1H, H-8); ^13^C NMR (151 MHz, D_2_O) δ 12.88 (C-18), 21.52, 24.10, 25.32, 27.39, 27.65, 27.84, 27.95, 27.98, 28.10, 30.60, 30.84 (C-7), 35.54 (C-3), 37.11 (C-2), 72.37 (C-10), 131.10 (C-9), 132.89 (C-8), 183.76 (C-1); ESI-HRMS: *m*/*z* [M-H]^−^ calcd. for C_18_H_33_O_3_: 297.2446, found: 297.2435; ESI-HRMS/MS data (Table 1).

### 2.5. Cell Line and Assessment for Antiproliferative Activity

All extracts, fractions, and compounds collected were analyzed for their antiproliferative activity using an MTT assay against human cervical cancer cells (HeLa) as described previously [2,11,12]. Cells were plated at a density of 1.5 × 10^3^ cells/well in 100 µL of EMEM and after 24 h they were treated with various concentrations of filter-sterilized extracts, fractions, or compounds. After another 48 h, cells were subjected to MTT assays and cell viability (%) was measured relative to the respective negative control used. For samples collected post-liquid–liquid extraction and Sephadex LH-20 column, concentrations at 0.5–1.0 mg/mL were prepared using methanol. For the linoleic acid, oleic acid, and linoleic acid autoxidation product (LAAP) mixture (for the sample at day 23 of autoxidation), cells were treated with samples prepared using methanol at concentrations of 400, 200, 100, 50, 25, 12.5, 6.25, and 3.125 µM. Doxorubicin was used as a positive control, and it showed approximately 80% antiproliferative activity. All doses were tested in triplicate in each experiment and the data shown are representative data from three biological replicates (*n* = 3).

### 2.6. Cell Line and Assessment for Immunomodulatory Activity

The RAW264.7 mouse macrophage cell line was purchased from the American Type Culture Collection (Rockville, MD, USA) and maintained in Dulbecco Modified Eagle Medium. Cells were grown in media supplemented with 10% fetal bovine serum in a humidified incubator at 37 °C, supplied with 5% carbon dioxide. To assess immunomodulatory activity, TNF-α production was measured in RAW264.7 cells upon treatment with 1 mg/mL fungal extracts for 6 h. Fifty microliters of medium were removed and stored at −80 °C until the TNF-α content was determined using the enzyme-linked immunosorbent assay (ELISA), as previously described [11,12]. 

### 2.7. NMR Analyses

The 1D and 2D NMR spectra were recorded on Bruker 600 MHz spectrometers, either with a 5 mm TXI or a TCI cryoprobe, using D_2_O as solvent. Standard Bruker pulse sequences were used for all data collections. Whenever needed, the residual water signal was suppressed with a low-power CW pulse during relaxation delays, or PE Watergate [13]. Chemical shifts (δ) are reported in ppm relative to TMS (δ = 0 ppm) and coupling constants (*J*) were reported in Hz. The following multiplicity of the ^1^H resonance peaks was used: singlet (s), doublet (d), triplet (t), quadruplet (q), and multiplet (m). Data processing was performed using MestReNova 14.2.1 (Mestrelab Research, Santiago de Compostela, Spain).

### 2.8. HPLC-MS and HPLC-MS/MS Analyses

HPLC-MS (electrospray ionization-low resolution mass spectrometry; ESI-LRMS) analyses were performed on an Agilent 1260 Infinity II Systems with a diode array detector coupled with an Agilent 6120 Single Quad MS. Analysis was carried out using a Phenomenex phenyl-hexyl column (4.6 mm × 250 mm × 5 µm), an Agilent Infinity Lab Poroshell 120 EC-C18 column (4.6 mm × 100 mm × 2.7 µm), and a Phenomenex Luna C18 (2) column (4.6 mm × 100 mm × 3 µm) with different gradient mobile phases as described above. ESI-LRMS spectra were obtained using an electrospray ion (ESI) source operated in positive or negative ion mode. The following ESI parameters were used: nebulizing gas N_2_ at 35 psig; drying gas N_2_ at 350 °C and 12 L/min; capillary voltage at 3000 V; and fragmentor voltage at 110 V. The MSD was operated in scan mode with the mass range of *m*/*z* 100–1000. 

HPLC-MS/MS (ESI-HRMS/MS) was carried out using an Agilent Infinite II 1260 LC system coupling with an Agilent 6545 QTOF MS system fitted with a Dual AJS ESI source. Agilent Mass Hunter version 10.0 was used for data acquisition. Separations were achieved using an Agilent Zorbax RRHD Eclipse Plus C18 column (2.1 mm × 50 mm × 1.8 µm) and an Agilent Infinity Lab Poroshell 120 EC-C18 column (2.1 mm × 50 mm × 2.7 µm) column with a mobile phase composed of an H_2_O solution of 0.1% formic acid (solvent A) and CH_3_CN containing 0.1% formic acid (solvent B) at a flow rate of 0.15–0.8 mL/min. Three different isocratic/gradient mobile phases were developed to achieve the necessary resolution and separation, and these were set as follows: (i) 0 min to 5 min (70% B, 0.15 mL/min); (ii) 0 min to 50 min (48% B, 0.5 mL/min); and (iii) 0 min (52% B, 0.8 mL/min), 23 min (52% B), and 25 min (90% B). ESI-HRMS or ESI-HRMS/MS spectra were acquired in negative ion mode. The following ESI parameters were used: nebulizing gas N_2_ at 20 psig; drying gas N_2_ at 325 °C and 10 L/min; capillary voltage at 4000 V; collision energies at 20–40 eV; and fragmentor voltage of 80/100 V. MS1 was operated in scan mode with the mass range of *m*/*z* 80–1100, scan rate at 10 spectra/s; the mass range of MS/MS mode was from 50 to 320 and the scan rate was 3 spectra/s.

### 2.9. Fatty Acid Methyl Esters Synthesis and GC-FID Quantification

Fatty acid composition analysis was performed at Lipid Analytical Laboratories Inc, Guelph, Ontario, Canada. The fatty acid compositions of all four of the studied mushrooms were determined according to the previously reported lipid extraction method [14], in the presence of known amounts of fatty acid methyl esters (FAMEs) as internal standards (Nu-Chek Prep Inc., Elysian, MN, USA) for total fatty acid analyses. An aliquot of the fatty acid extracted was used to form FAMEs through transmethylation following a previously published method [15]. The FAMEs were prepared using boron trichloride in methanol and the methylation tubes were heated to 95 °C in a boiling water bath. The FAMEs were analyzed on an Agilent 7890B GLC (Santa Clara, CA, USA) equipped with a flame ionization detector (FID) and a 60 m DB-23 capillary column (0.32 mm internal diameter) [16]. FAMEs were identified using a standard mixture (qualitative and quantitative) with 49 known fatty acid components for verification, which were obtained from the American Oil Chemists Society (Champaign, IL, USA) and Sigma-Aldrich (St. Louis, MO, USA).

### 2.10. Autoxidation of Linoleic Acid

Linoleic acid solution (32 mM, 2.4 mL) was prepared in methanol and distributed into six different tubes (400 µL each). All the samples were left uncapped to oxidize at room temperature (22 °C). One sample tube was used each time and diluted with methanol before subjected to HPLC analysis on days 0, 2, 4, 10, 15, and 23. All data were obtained from triplicates for each sample and are expressed as means ± S.D. from three independent experiments.

## 3. Results

### 3.1. Assessment of Crude Extracts for Immunomodulatory and Antiproliferative Activities

Powdered *O. tomentosa* (CL83) was sequentially extracted with 80% ethanol (E1), 50% methanol (E2), water (E3), and 5% NaOH (E4) and assessed for antiproliferative and immunomodulatory activities, as shown in Figure 1. The dose-dependent (Figure 1a) MTT assay showed that E2 and E3 from *O. tomentosa* only had a weak antiproliferative effect (20–25% inhibition) on HeLa cells, whereas E4 exhibited mediocre activity of up to 50% inhibition at 0.25 mg/mL (Figure 1a). In contrast, E1 from *O. tomentosa* showed a strong antiproliferative activity with up to 90% inhibition at 1 mg/mL (Figure 1a). Therefore, E1 was chosen for further studies. To assess the extent of E1’s inhibitory effect, an intermediate dose of 0.5 mg/mL was added to cells for 7 days. As shown in the time-dependent MTT assay (Figure 1b), as compared to the control, the inhibitory effect of E1 persisted for up to 7 days after treatment. The crude extracts from *O. tomentosa* were also assessed for potential immunomodulatory activity and the results are shown in Figure 1c. E1, E2, and E4 displayed no immunomodulatory activity. The water extract E3 exhibited somewhat weak activity compared to the positive control lipopolysaccharide (Figure 1c), as well as to extracts from other mushroom species that we had previously investigated [11,12].

### 3.2. Bioactivity-Guided Fractionation and Purification

To confirm the antiproliferative activity of the E1 extract from *O. tomentosa*, an extract prepared from a different collection of *O. tomentosa* CL312 was assessed. As before, the 80% ethanol extract (E1) from *O. tomentosa* CL312 displayed stronger antiproliferative activity as compared to the 50% methanol extract (E2) (Appendix A). As shown in Appendix A, at 1 mg/mL, extract E2 showed only 50% inhibition, whereas extract E1 exhibited 80% inhibition. Based on this finding, we focused in this study on isolating antiproliferative compounds from the E1 extract.

Extract E1 was first dissolved in water and subjected to sequential extraction using organic solvents with increasing polarity via the liquid–liquid extraction method. The results showed that all the organic layers, hexane (HEX), diethyl ether (DEE), and chloroform (CHCl_3_), exerted a growth inhibition of approximately 77–85% on HeLa cells at a concentration of 0.5 mg/mL (Appendix A). In contrast, the water layer was relatively inactive, showing only 14% inhibition at 1 mg/mL. These results suggested that most bioactive compounds were present in the organic layers. HPLC-UV analysis was carried out on these four organic layers using a diode array detector (DAD) set at different wavelengths of 200 nm (Appendix A), 230 nm (Appendix A), and 280 nm (Appendix A). To our surprise, at λ = 200 nm (Appendix A) several chromatographic peaks were found with retention time (RT) between 23 and 30 min in all four organic layers, suggesting that small molecules detected in this time range and wavelength could be correlated with the previously observed antiproliferative activity (Appendix A).

Since all the organic layers showed antiproliferative activity and most likely contained the same small molecules (RT 23–30 min), only HEX and DEE layers were chosen for further purification using a Sephadex LH-20 column. Out of the 36 fractions collected for the HEX layer, fractions 10 to 20 showed more than 70% inhibitory activity (Appendix A). For the DEE layer (Appendix A), bioactivity was detected at the same fractions (fractions 10–20). Thus, HEX and DEE solvents appeared to have extracted similar small molecules, which is in good agreement with our previous HPLC analysis. Subsequently, HPLC analysis was performed on the active fractions collected (Appendix A) at selected wavelengths (200, 230, 250, and 280 nm). Before Sephadex LH-20 purification (Appendix A), most of the peaks were detected between RT of 23.4–29.3 min and some other peaks at RT of 5.1, 7.2, 12.6, and 14.4 min. It is expected that after Sephadex LH-20 purification, fewer peaks would be detected. However, to our surprise, several new peaks which were not previously detected appeared at the RT of 16–20 min (Appendix A) after the Sephadex LH-20 purification step. In comparing the HPLC spectra before and after Sephadex LH-20 purification (Appendix A), these new peaks showed absorbance at different wavelengths, which could be used later for detection purposes. In summary, peaks at RT of 23.4–29.3 min were detected at a wavelength of 200 nm, whereas newly formed peaks at RT of 16.7–17.5 min and RT of 17.8–18.6 min were detected at 230 nm and 280 nm, respectively. Hence, these three wavelengths were selected for subsequent purification and analysis.

To better understand which compounds the new peaks that appeared post-Sephadex LH-20 purification might represent, samples were analyzed via ESI-LRMS (Appendix A) using a newly optimized chromatographic protocol with better separation and resolution. The HPLC method was optimized accordingly for each sample throughout the study to achieve the best chromatographic resolution. A total of seven compounds were focused and labeled as compounds **1**–**7** as shown in Appendix A. Small molecules with molecular weights of 296 (**1**), 294 (**2–4**), 298 (**5**), 280 (**6**), and 282 (**7**) Da (found as peaks with *m*/*z* 295, 293, 297, 279, and 281, respectively) were detected using the negative ion ESI mode. 

Subsequently, purification was performed on these compounds using an HPLC reverse-phase column with two separate chromatographic separation protocols (Appendix A). Mixture **1** (Appendix A) consisted of two major overlapping peaks with the same molecular weight of 296 Da, inseparable even with different chromatographic separation protocols. They were analyzed as a mixture in this study. Mixture **1** and compounds **3**, **4**, and **5** were collected (Appendix A) and further analyzed post-HPLC purification (Appendix A). Surprisingly, mixture **1** separated into two peaks at RT of 10.7 min and 12.7 min post-HPLC purification (Appendix A). These two peaks (RT = 10.7 min and 12.7 min) were subsequently re-fractionated using an optimized method (Appendix A) and were found to be unstable and interchangeable from one form to the other, giving two major (compounds **1.3** and **1.4**) and two minor peaks (compounds **1.1** and **1.2**). ESI-LRMS analysis of these four peaks revealed that they all have the same parent ion of *m*/*z* 279 [M-H_2_O+H]^+^, suggesting the presence of four peaks in mixture **1**, which were later named as compounds **1.1**–**1.4** in this study. Based on this finding, no further purification was performed on compound **2**, which also showed as two overlapping peaks. For compound **3**, only a single peak was detected (Appendix A) after purification. For compound **4**, two separate peaks at RT 17.1 min (**4**) and 20.3 min (**5**) were seen after the purification (Appendix A). It was found that some compound **4** was degraded to form **5**. Conversely, compound **5** (Appendix A) remained as a single peak with compound **4** being undetectable. Appendix A shows the HPLC reversed-phase column purification results for compounds **6** and **7**. For compound **7**, only a major peak was detected (Appendix A) after purification. However, for compound **6** (Appendix A), several new peaks between RT of 11.2–13.5 min were detected at 200 nm. Subsequent HPLC-MS analysis on compound **6** (Appendix A) detected several peaks between RT of 10.9–16.6 min with molecular weights ranging from 294 to 298 Da. The mass detected was the same as that of mixtures **1.1**–**1.4**, and compounds **2**–**5**, suggesting that the appearance of these compounds (**1.1**–**1.4** and **2**–**5**) post-Sephadex LH-20 purification was most likely the degradation product of compound **6**. All compounds (**1.1**–**1.4** and **2**–**7**) were further analyzed via HPLC-MS/MS analysis. Due to the low abundance of some of the unstable compounds, only compound **5** was submitted for NMR analysis.

### 3.3. Identification of Compounds

Compound **5** was identified as 10-hydroxy-(8*E*)-octadecenoic acid by comparing its NMR spectroscopic data (Appendix A) with those previously published [17,18,19]. The ^1^H NMR spectrum (Appendix A) disclosed one terminal fatty acid methyl hydrogen signal at 0.86 ppm (H-18), 13 methylene hydrogen signals at 1.21–2.17 ppm (H-2 to H-7 and H-11 to H-17), one hydrogen on the hydroxyl carbon signal at 4.08 ppm (H-10), and olefinic proton signals at 5.46 ppm (H-9) and 5.70 ppm (H-8). The ^13^C NMR spectrum (Appendix A) exhibited 18 carbons that were attributed to one methyl carbon at 12.88 ppm (C-18), 13 methylene carbons at 21.5–37.1 ppm (C-2 to C-7 and C-11 to C-17), one hydroxyl carbon at 72.37 ppm (C-10), two methine/olefinic carbons at 131.10 (C-9) and 132.89 ppm (C-8), and a carbonyl carbon at 183.76 ppm (C-1). ^1^H-^1^H COSY (Appendix A) showed that the H-10 on the hydroxyl carbon at 4.08 ppm was correlated with the H-8 and H-9 olefinic protons at 5.46 (dd, *J* = 15.4, 7.5 Hz) and 5.70 ppm (dt, *J* = 15.5, 6.9 Hz). This indicates that the hydroxyl group was located at the neighboring position of the double bond. The coupling constant of the double bond at 15.4 Hz strongly suggests that it was in a *trans* configuration. The *trans* configuration was further substantiated by ^1^H NOESY data analysis. As shown in Appendix A, a strong NOE between H-8 and H-10 was observed, but no or negligible NOE was observed between H-8 and H-9. If it were a *cis* configuration, a strong NOE between H-8 and H-9 and a weak NOE between H-8 and H-10 would have been observed.

Compound **5** was further confirmed by the ESI-HRMS/MS data. The mass of the precursor ion [M-H]^−^ (*m*/*z* 297) in the HRMS spectrum (Appendix A) matched well with the expected molecular formula of C_18_H_34_O_3_. The precursor ion decomposed and produced abundant *m*/*z* 279 [M-H_2_O-H]^−^ in the MS/MS spectrum (Appendix A), which corresponded to the loss of a water molecule, indicating that it is a hydroxy-diene fatty acid instead of a keto-diene. It also exhibited a peak at *m*/*z* 155 [M-C_9_H_18_O-H]^−^ which was caused by bond cleavage adjacent to the hydroxyl group, as shown in Figure 2, and coinciding with the reported literature data [20]. Compound **5** is a fatty acid based on the NMR spectroscopic data and fragmentation profiles obtained via ESI-HRMS/MS. As mentioned above, mixture **1** (**1.1**–**1.4**) and compounds **2** to **5** have molecular weights of 294, 296, or 298 Da, suggesting that they only differed by the degrees of unsaturation in their structure, indicating they are most likely to be fatty acid derivatives.

All compounds in mixture **1** (compounds **1.1** to **1.4**) showed parent ions at *m*/*z* 296 for [M-H]^−^ (Appendix A), indicating that it was a mixture of isomers with a molecular formula of C_18_H_32_O_3_. Dehydration of compounds **1.1** to **1.4** led to ion detection at *m*/*z* 277 [M-H_2_O-H]^−^ (Table 1), indicating that they were all hydroxy-diene fatty acids. The MS/MS spectrum of compound **1.1** (Appendix A) showed strong signals at *m*/*z* 157 [M-C_10_H_18_-H]^−^. The literature points out that *m*/*z* 157 is a characteristic ion formed by α-cleavage between C8 and C9 bonds (Figure 2) with the transfer of a proton to the unsaturated side for 8-hydroxy conjugated fatty acid. Thus, comparing the MS/MS spectrum with those reported in the literature confirmed compound **1.1** as 8-hydroxy-(9*Z,* 12*Z*)-octadecenoic acid [21,22]. The MS/MS spectrum of compound **1.2** showed main fragment ions at *m*/*z* 183.1390 [M-C_7_H_12_O-H]^−^ (Appendix A) resulting from the cleavage between C11 and C12 (Figure 2). Compound **1.3** also showed an ion at *m*/*z* 183. However, MS/MS showed an accurate mass of *m*/*z* 183.1026 [M-C_8_H_16_-H]^−^ (Appendix A) which is slightly different from the *m*/*z* 183.1390 of compound **1.2**. This indicates cleavage between different carbons (C10 and C11), instead of C11 and C12, as shown for compound **1.2** (Figure 2). Compounds **1.2** and **1.3** also contained fragment ions at *m*/*z* 211 and 155, respectively (Figure 2), which were caused by bond cleavage between the diene and hydroxyl group. Comparison of fragment profiles with literature data allowed the identification of compounds **1.2** and **1.3** as 12-hydroxy-(9*Z,* 13*E*)-octadecenoic acid and 10-hydroxy-(8*E,* 12*Z*)-octadecenoic acid, respectively [20,23]. For compound **1.4**, Two characteristic product ions were detected at *m*/*z* 171 [M-C_9_H_16_-H]^−^ and 123 [M-C_9_H_16_O_3_-H]^−^ (Appendix A), resulting from the fission between C9 and C10 (Figure 2). The comparison of the MS/MS spectra and UV absorption wavelength (230 nm) with literature data led to the assignment of compound **1.4** as 9-hydroxy-(10*E*, 12*E*/*Z*)-octadecadienoic acid [20,23,24].

Compounds **2** and **3** have molecular weights of 294 Da as shown by the deprotonated parent ion with *m*/*z* 293 [M-H]^−^ (Appendix A), fitting well with the molecular formula of C_18_H_30_O_3_. They both displayed a product ion of *m*/*z* 249 [M-CO_2_-H]^−^ (Appendix A) resulting from the loss of CO_2_, indicating that they are keto-dienes linoleic acid derivatives. Compounds **2** and **3** showed similar MS/MS fragmentation ions (Table 1), suggesting that the difference between them is based on the stereochemistry of the double bond. They both displayed fragment ions of *m*/*z* 195 [M-C_6_H_10_O-H]^−^ resulting from the cleavage between C12 and C13 (Figure 2) and *m*/*z* 179 [M-C_7_H_14_O-H]^−^ (Appendix A), which are important characteristic ions of 13-oxo conjugated fatty acid. Subsequent fragmentation of the *m*/*z* 195 ion showed the production of ion at *m*/*z* 167 [M-C_8_H_14_O-H]^−^, resulting from the loss of ethylene [24]. They also displayed a base peak at *m*/*z* 113 [M-C_11_H_16_O_2_-H]^−^ from the cleavage between C11 and C12 (Figure 2) [20]. Combining all this information, compounds **2**/**3** were determined to be 13-oxo-(9*Z*, 11*E*)-octadecadienoic acid and 13-oxo-(9*E*,11*E*)-octadecadienoic acid.

Compound **4** showed the same parent ion *m*/*z* 293 [M-H]^−^ and product ion *m*/*z* 249 [M-CO_2_-H]^−^, as compounds **2**/**3** (Appendix A). However, it showed a different set of fragment ions at *m*/*z* 197 [M-C_7_H_12_-H]^−^, 185 [M-C_8_H_12_-H]^−^, and 125 [M-C_10_H_16_O_2_-H]^−^ (Table 1), which are important diagnostics ions for 9-ketodienes [24]. The main fragments present in the MS/MS spectrum are consistent with previously published data [20,24]. Thus, the structure of compound **4** was elucidated to be 9-oxo-(10*E*, 12*E*/*Z*)-octadecadienoic acid.

Compounds **6** and **7** showed parent ions of *m*/*z* 279 [M-H]^−^ and *m*/*z* 281 [M-H]^−^ (Appendix A), suggesting a molecular formula of C_18_H_32_O_2_ and C_18_H_34_O_2_, respectively. Linoleic (280 Da) and oleic acids (282 Da) have been previously identified as the main fatty acids found in different mushroom species [25,26,27]. Hence, compounds **6** and **7** were identified by matching their specific HPLC retention times and mass fragmentation (MS/MS) with reference to the commercially purchased pure standard, which was run under the same chromatographic conditions. Comparison of the MS/MS analysis of compound **6** (Appendix A) with linoleic acid (Appendix A) and compound **7** (Appendix A) with oleic acid (Appendix A) showed identical fragmentation profiles to those of the two standards. Subsequent HPLC analysis showed that the retention time of compounds **6** (RT = 30.0 min) and **7** (RT = 32.8 min) matched the retention time of the two pure standards (Appendix A). Thus, compounds **6** and **7** were identified as linoleic and oleic acid, respectively.

### 3.4. Autoxidation/Degradation of Linoleic Acid

To understand how the autoxidation of fatty acids occurred during their isolation from *O. tomentosa*, the disappearance of linoleic acid (Figure 3) and the formation of its oxidation products, hydroxyoctadecadienoic acids (HODEs) and oxooctadecadienoic acids (KODEs) (Figure 4) were simultaneously measured at different wavelengths using HPLC. Linoleic acid was detected at 200 nm, whereas HODEs and KODEs were detected at 230 nm and 280 nm, respectively. Similar wavelengths have also been used in other studies [24,28].

During the first four days, the linoleic acid oxidation level remained low (Figure 3). After day 4, the level of linoleic acid decreased significantly and reached a minimum level (0.4 mM, 1.2%) on day 23. Conversely, mixtures **1** (**1.1**–**1.4**, also named as 8-, 12-, 10-, and 9-HODEs) and compounds **2**–**4** (13- and 9- KODEs) increased rapidly starting on day 4 and reached maximum levels on day 10 (Figure 4), supporting the notion that linoleic acid was converted to HODEs and KODEs, as observed during our isolation of bioactive compounds from *O. tomentosa*. 

### 3.5. Fatty Acid Composition in Selected Mushrooms

Since fatty acids were the major antiproliferative compounds detected in *O. tomentosa*, it was of interest to determine the fatty acid composition of *O tomentosa* and to compare it with those of other collected mushrooms that have not been previously studied for their fatty acid contents. These include *F. officinalis*, *E. tinctorium* and *A. flettii*. A total of 35 fatty acids with carbon chains of C_11_ to C_24_ were identified in the four mushroom species (Table 2). The short-chain fatty acids from C_6_ to C_11_ were not detected in any of the four mushroom samples. Saturated fatty acids (SFAs), monounsaturated fatty acids (MUFAs), and polyunsaturated fatty acids (PUFAs) are reported as percentages (%) of total fatty acids (Table 2) and micrograms of fatty acids determined per gram of dried mushroom samples (µg/g) (Appendix A).

The lipid content was reported to be 0.5–3.0% per 100 g of four dried mushrooms studied (Appendix A). However, the lipid content of *E. tinctorium* was 0.5%, which is lower than the general lipid content range (1.75–15.5%) of various other dried mushroom species [29]. Linoleic (18:2n6; 15.9–51.7%), oleic (18:1; 22.8–57.1%), palmitic (16:0; 6.0–14.6%), and stearic (18:0; 3.7–10.4%) acids were the major fatty acids found in all four mushroom species. We should also point out that *A. flettii* (51.7%) and *O. tomentosa* (48.4%) showed a higher content of linoleic acid as compared to *F. officinalis* (17.1%) and *E. tinctorium* (15.9%). Despite having a lower linoleic acid content, *F. officinalis* (57.1%) and *E. tinctorium* (34.7%) showed a higher oleic acid content as compared to *A. flettii* (22.8%) and *O. tomentosa* (23.6%). Moreover, pentadecylic (C15:0, 0.3–2.1%), palmitoleic (C16:1, 0.3–2.1%), heptadecanoic (C17:0, 0.3–2.2%), vaccenic (C18:1, 1.5–2.8%), γ-linoleic (C18:3n6, 0.1–0.5%), arachidic (C20:0, 0.3–0.9%), docosahexaenoic acid (C22:6n3, 0.4–2.3%) were also detected, but in a lower percentage, in all studied mushrooms.

### 3.6. Antiproliferative Activity of Fatty Acids

Linoleic and oleic acids were detected as the major components in the antiproliferative fractions from *O. tomentosa* (Appendix A). To determine whether they were one of the major antiproliferative compounds in the bioactive fractions, these compounds were directly assessed for their antiproliferative activity against HeLa cells. Figure 5 shows that both linoleic and oleic acids were antiproliferative, with IC_50_ values of 50.3 ± 10.8 µM and 90.4 ± 3.1 µM, respectively. Considering that some of the fatty acids detected in the bioactive fractions were the autoxidation products of linoleic acid, the cytotoxicity of the linoleic acid autoxidation product (LAAP) mixture (for the sample at day 23 of autoxidation) was also assessed. Surprisingly, LAAP also showed inhibitory activity, with IC_50_ value of 42.8 ± 7.1 µM (Figure 5). Doxorubicin, used as the positive control, showed an IC_50_ value of 0.36 ± 0.17 µM against HeLa cells (Appendix A). Therefore, as compared to doxorubicin, the antiproliferative activities of linoleic acid, oleic acid, and LAAP were less potent.

## 4. Discussion

This study aimed to investigate the potential medicinal properties of the polypore fungus *O. tomentosa*, which is found throughout the forests in north–central BC. Its ethanol extract exhibited strong antiproliferative activity, whereas its water extract exhibited weak immunomodulatory activity. The use of the liquid–liquid extraction method and Sephadex LH-20 purification coupled with HPLC-MS, followed by HPLC-MS/MS and NMR, led to the identification of eight known linoleic oxygenated fatty acids (**1.1**–**1.4** and **2**–**5**), together with linoleic acid (**6**) and oleic acid (**7**), which were the major antiproliferative constituents in the ethanol extract of *O. tomentosa*. 

Linoleic acid is prone to oxidation, resulting in bioactive linoleic acid derivatives such as hydroxyoctadecadienoic acids (HODEs). It can be first oxidized to hydroperoxy-octadecadienoic acids (HPODEs) through three possible mechanisms: free radical-mediated oxidation, free radical-independent nonenzymatic oxidation (which generates ^1^O_2_), and enzymatic oxidation. HPODEs are then reduced by several enzymes to form HODEs [30]. Compound **1.1** (also known as 8-HODE) was previously reported to be the oxidation product of linoleic acid by the linoleate diol synthase (LDS) enzyme [22]. Compounds **1.2** (12-HODE), **1.3** (10-HODE), and **1.4** (9-HODE) were previously reported to be generated via non-radical (singlet oxygen-mediated) oxidation of linoleic acid through the treatment of cells with ^1^O_2_ during UV-A irradiation [30]. For the formation of compounds **2**–**4**, linoleic acid is first oxidized through radical-mediated oxidation to form 9- and 13-hydroperoxy-octadecadienoic acid (9-HPODE and 13-HPODE), which are then subsequently reduced to form 9-HODE (compound **1.4**) and 13-HODE (not detected in this study). Both 9-HODE (compound **1.4**) and 13-HODE can be further oxidized by hydroxy-fatty acid dehydrogenase to form 9- and 13-oxo-octadecadienoic acid (9-KODE, compound **4**; 13-KODE, compound **2**/**3**) in vivo [31]. Unlike mixture **1** (**1.1**–**1.4**) and compounds **2** to **4**, compound **5** has been identified as a product of the microbial oxidation of oleic acid in several studies [32,33,34,35,36]. *Pseudomonas* sp. 42A2 is known to convert oleic acid to produce 10-hydroxy-8*E*-octadecenoic acid (compound **5**) and 7,10-dihydroxy-8*E*-octadecenoic acid through microbial oxidation [32,33]. The same result has also been reported when other *Pseudomonas* strains were used [34,35,36]. In this study, we found that compound **4** was converted to compound **5** after the purification step (Appendix A), suggesting that compound **5** was an oxidized product of compound **4**. Hence, we report here for the first time that compound **5** was a product of the autoxidation of linoleic acid through the oxidation of compound **4**.

Several studies have shown that linoleic acid derivatives identified in this study can be generated via the incubation of linoleic acid with lamellae [37], mycelial mats [38], and LDS enzymes [22] extracted from different fungi. In these studies, oxygenated fatty acids and linoleic acid derivatives were classified as fungal fatty acid metabolism products. The present study on *O. tomentosa* is the first report of the autoxidation of linoleic acid upon its isolation from fungi. 

Interestingly, all linoleic oxygenated fatty acids detected in this study (compounds **1.2**–**1.4** and **2**–**5**), with the exception of compound **1.1,** were found in the plant *Alternanthera brasiliana* during the isolation of antibacterial compounds [20]. Although linoleic acid was not detected in the antibacterial fractions, the authors speculated that the antibacterial fatty acids from *A. brasiliana* were derived from the interaction between the plant and the endophytic bacteria, which was later found to produce compounds **1.2**–**1.4**. Linoleic acid, which is abundant in plants, was likely to undergo autoxidation, resulting in a large number of linoleic acid derivatives. The present study clearly demonstrated the instability of fatty acids in mushrooms and the challenges that researchers will encounter when isolating linoleic acid-type biomolecules from a natural source.

In this study, pure linoleic acid was shown to autoxidize to HODEs and KODEs (Figure 4). The formation of HODEs and KODEs has also been reported using BSA-bound linoleic acid, where the concentration of HPODEs, HODEs, and KODEs increased following the degradation of linoleic acid [24]. In another study, it was reported that during the autoxidation of linoleic acid, HPODEs showed a maximum increase up to 12–16 h, followed by a decrease. HPODEs have been reported to be unstable and can produce radicals that can initiate or propagate the autoxidation steps. KODEs, the degradation product of HPODEs, showed an increase throughout the 24 h of oxidation time [28]. HODEs and KODEs are less stable due to the presence of two double bonds which are susceptible to oxidation. In the present study with longer oxidation times (after day 10, Figure 4), both HODEs and KODEs were further degraded to secondary products, which were detected via MS but not via DAD (data not shown). The formation of these secondary products suggested that products other than HODEs and KODEs were formed.

This study convincingly showed that linoleic acid was inhibitory on the viability of HeLa cells with an IC_50_ of 50.3 µM (Figure 5). However, an earlier study showed that linoleic acid possessed a growth-stimulatory effect on HeLa cells [39]. A closer look at the published paper revealed significant differences between our study and the work conducted by Sagar and Das [39]. Firstly, RPMI medium supplemented with human albumin serum was used in their study, as opposed to the use of EMEM with fetal calf serum in this study. Secondly, the authors used the trypan blue exclusion method to count the number of cells and ^3^H-thymidine incorporation to measure DNA synthesis [39]. In contrast, we used the MTT assay in this study to measure cell viability. Oddly, the authors showed that linoleic acid was growth-stimulatory at 10 µg/mL, had no significant effect at 20 µg/mL, and was only stimulatory 48 h after treatment with linoleic acid at 40 µg/mL [39]. Therefore, the discrepancy between the results from this study and those of Sagar and Das on HeLa cells could be due to the different culture conditions, the types of assays, the concentration of linoleic acid used, or the combination of all three. Interestingly, linoleic acid has been shown to be both growth-inhibitory and growth-stimulatory, depending on cell types and the concentrations used. For instance, it was growth-inhibitory against human colorectal carcinoma cancer cells (Caco-2) at 160 µM [40], human prostate cancer cells (PC-3), and human prostate epithelial (RWPE-1) cells at the range of 100–200 µM [41]. However, a lower concentration of linoleic acid (25 µM) was found to promote the growth of both PC-3 and RWPE-1 cells [41]. Another study showed that linoleic acid was found to enhance cellular proliferation in human breast ductal carcinoma cells (BT-474) and human lung adenocarcinoma cells (A549) when a higher concentration (50 µM) was used [42]. In summary, linoleic acid appears to have both growth-stimulatory effects and growth-inhibitory depending on the types of cells, the culture conditions, and the concentration used.

Oleic acid showed antiproliferative activity, with an IC_50_ value of 90.4 ± 3.1 µM (*n* = 3). This result is consistent with the previously reported cytotoxicity of oleic acid, which showed less than 50% cell viability at a tested concentration of 100 µM [43]. In addition to HeLa cells, oleic acid has been shown to inhibit the proliferation of breast cancer cells (MCF-7) and human colorectal adenocarcinoma cells (HT-29) with IC_50_ values of 162 µg/mL and 50 µg/mL, respectively [44]. However, other studies have shown that oleic acid promoted the proliferation of human prostate cancer cells (PC3) at 100–400 µM [45] and Caco-2 cell growth at 1–100 µM [46]. 

The results on the analysis of the fatty acid composition of four mushrooms native to north–central BC, including *O. tomentosa*, are consistent with studies conducted by others on different mushrooms in which linoleic, oleic, and palmitic acid were found to account for most of the fatty acids detected [25,26,27]. A literature search revealed that linoleic acid is most abundant in mushrooms from Oceania (59.4 g/100 g), followed by Europe (48.6 g/100 g) and Africa (48.5 g/100 g), whereas oleic acid is the second most abundant [47]. Linoleic acid is a PUFA and belongs to the omega 6 (ω-6) family, whereas oleic acid is a MUFA and belongs to the omega 9 (ω-9) family. They were the two most dominant UFAs, whereas palmitic and stearic acid were the most dominant among SFAs in the four studied mushroom species. The percentages of SFAs, MUFAs, and PUFAs ranged from 13.2% to 31.9%, 27.6% to 65.5%, and 21.3% to 54.3%, respectively, among the four mushroom species. Furthermore, the total content of UFAs was higher than that of SFAs in all four studied mushroom species, mainly due to the high abundance of linoleic and oleic acids. The highest amount of UFAs was found in *F. officinalis* (86.8%), followed by *A. flettii* (81.9%), *O. tomentosa* (78.3%), and *E. tinctorium* (68.1%). *O. tomentosa* is closely related to *I. leporinus* and *I. circinatus* [48,49,50]; however, neither species had their fatty acids composition analyzed. Amongst the *Inonotus* species, the fatty acid contents were reported in *I. hispidus* [51], *I. obliquus* [52], and *I. radiatus* [51]. Linoleic and palmitic acids were the major fatty acids found in *I. obliquus* [52] (Appendix A). Conversely, *I. hispidus* showed a different fatty acid profile with arachinic, oleic, and pentadecanoic acids as the major fatty acids (Appendix A). *O. tomentosa* has never been explored as a possible edible or medicinal mushroom. Therefore, the results regarding its fatty acid content and activity reported here represent valuable information for future investigations in this area of study.

In recent years, fatty acids have been used in the characterization and classification of unknown mushroom species, along with mathematical, statistical, and other logic-based methods, as an alternative to conventional morphological identification [26,27,53]. By applying multivariate statistical analysis methods to the study of fatty acid composition, it was shown that fatty acid composition could be used to discriminate between two types of families of mushrooms [27]. For instance, significant differences were observed in the score plots of *Agaricaceae* (positive score) and *Tricholomataceae* (negative score) mushrooms due to the differing contents of PUFAs and SFAs in the two fungi [27]. To this end, the results reported in this study are expected to aid such endeavors further.

## 5. Conclusions

Antiproliferative activity-guided fractionation and purification of the ethanol extract of *O. tomentosa*, in combination with the ESI-HRMS/MS and NMR experiments, led to the identification of eight known linoleic oxygenated fatty acids (**1.1**–**1.4** and **2**–**5**), together with linoleic acid (**6**) and oleic acid (**7**). Diode array UV detection aided the identification of compounds **1.1**–**1.4** (HODEs) at 230 nm, compounds **2**–**4** (KODEs) at 280 nm, compound **5** at 280 nm, and compounds **6** and **7** at 200 nm. The oxidation of linoleic acid between positions C9 and C13 produced compounds **1.1**–**1.4** and **2**–**5**, which were further degraded to secondary products with a longer oxidation time. Although these compounds (**1.1**–**1.4** and **2**–**5**) have been previously reported, compound **5** is reported here for the first time as an autoxidation product of linoleic acid. GC-FID was used to determine the composition and abundance of fatty acids in four selected mushrooms native to north–central BC, including *O. tomentosa*. Linoleic (15.9–51.7%), oleic (22.8–57.1%), palmitic (6.0–14.6%), and stearic (3.7–10.4%) acids were found to be the major fatty acids in *O. tomentosa*, *A. flettii*, *E. tinctorium*, and *F. officinalis*. This study showed that linoleic acid can undergo autoxidation upon its isolation from a natural source, indicating the need for caution in regard to the challenges involved in isolating unstable fatty acids, such as linoleic acid, from natural sources.

## Figures and Tables

**Figure 1 jof-08-01163-f001:**
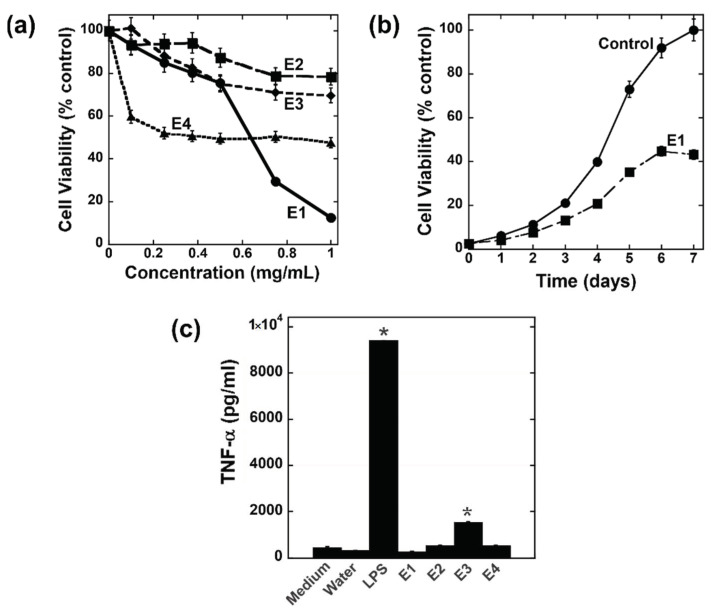
Antiproliferative and immunomodulatory activities of crude extracts from *O. tomentosa*. Dose- (**a**) and time-dependent (**b**) MTT cell viability assays on HeLa cells. The concentration of E1 used for the time-dependent experiment (**b**) was 0.5 mg/mL. (**c**) At 1 mg/mL, crude extracts from *O. tomentosa* were assessed for their ability to induce TNF-α production in RAW264.7 macrophage cells as an indicator of immunomodulation. Lipopolysaccharide (LPS) was used as a positive control, whereas medium and water were used as negative controls. The results presented are representations of two separate experiments *(n* = 2). Error bars are SD. One-way ANOVA (Tukey Test) was used for statistical analysis. * denotes *p* < 0.05 as compared to the water control.

**Figure 2 jof-08-01163-f002:**
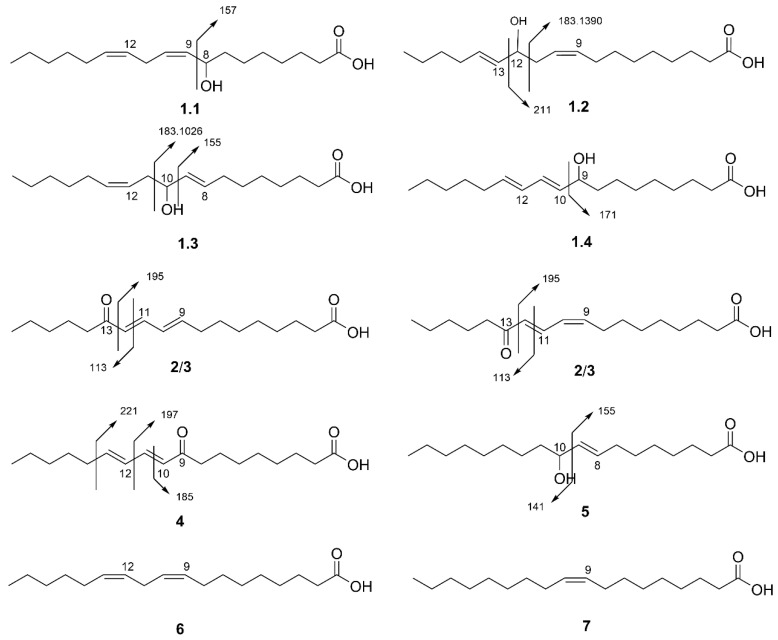
Molecular structures and selected MS/MS fragments of mixture **1** (**1.1**–**1.4**) and compounds (**2**–**7**) found in *O. tomentosa* antiproliferative fractions.

**Figure 3 jof-08-01163-f003:**
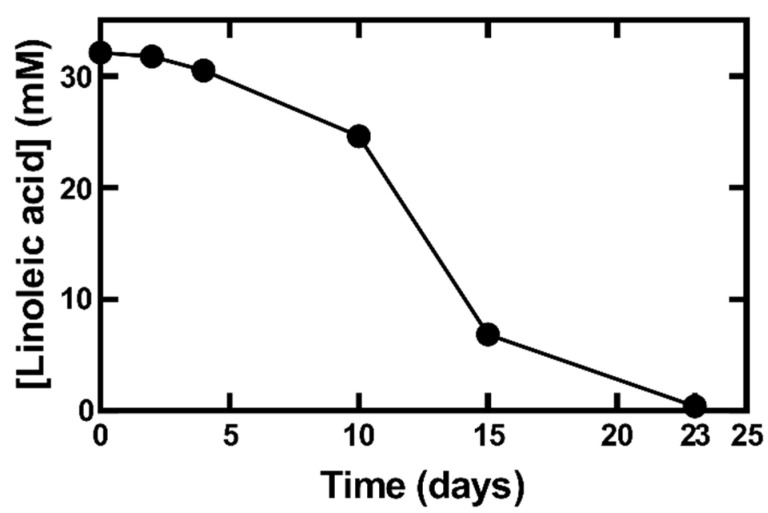
Autoxidation of linoleic acid. Linoleic acid was incubated at room temperature, and on different days an aliquot was taken for quantification via HPLC. Linoleic acid on days 2, 4, 10, 15, and 23 was quantified by comparing the peak area of products relative to day 0 (32 mM). The results shown (mean ± S.D.) were averaged from three replicates (*n* = 3).

**Figure 4 jof-08-01163-f004:**
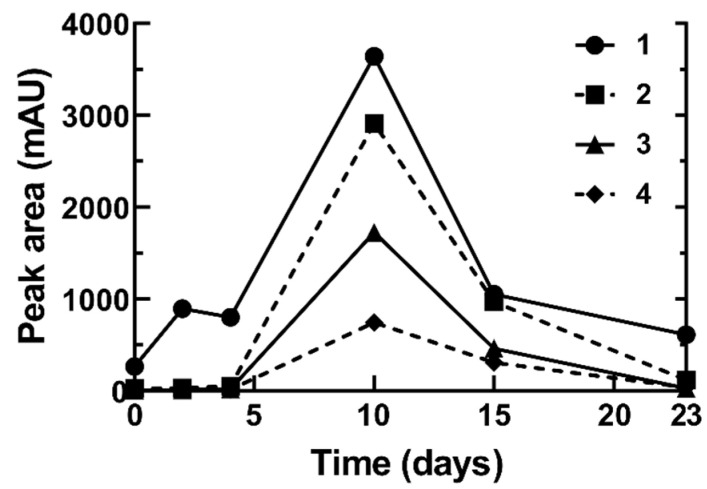
Formation of HODEs and KODEs upon autoxidation of linoleic acid at room temperature. Separation of the mixture **1** (HODEs) peak could not be fully achieved, and it was analyzed as a mixture. The peaks identified were as follows: **1**: mixtures of 8-HODE, 9-HODE, 10-HODE, and 12-HODE; **2**/**3**: 13-KODE; **4**: 9-KODE. All HODEs and KODEs were quantified by comparing the peak areas of products relative to day 0. HPLC was used to monitor the autoxidation reactions and the results were shown as means ± S.D., averaged from three replicates (*n* = 3).

**Figure 5 jof-08-01163-f005:**
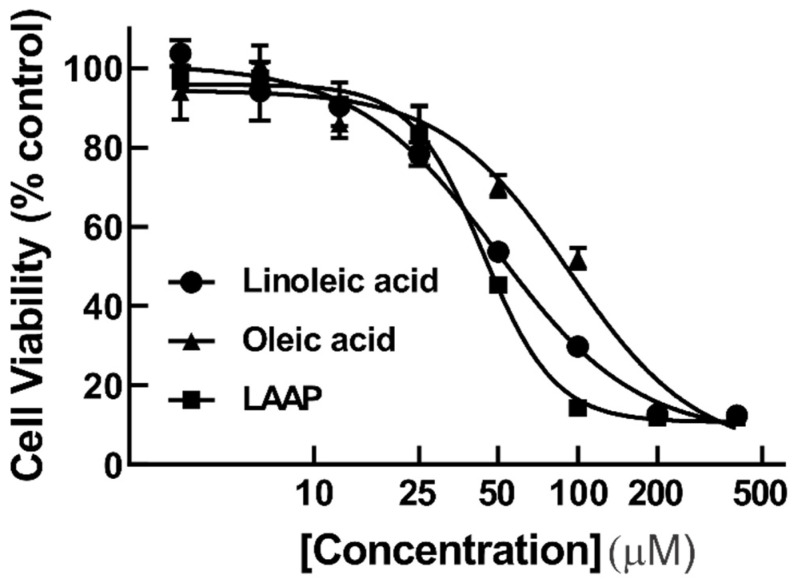
The linoleic acid, oleic acid, and LAAP mixture exhibited antiproliferative activity against HeLa cells. HeLa cells were treated with the linoleic acid, oleic acid, and LAAP mixture for 48 h at 400, 200, 100, 50, 25, 12.5, 6.25, and 3.125 µM. Cell viability was determined using the MTT assay. The result shown is representative data from three biological replicates (*n* = 3). Error bars indicate standard deviation (S.D.).

**Table 1 jof-08-01163-t001:** HRMS (MS and MS/MS) analysis of fatty acids from *O. tomentosa*.

Compound	Molecular Formula	Precursor Ion [M-H]^−^	ppm Error	Fragment Ions [M-H]^−^ by MS/MS ^a^
**1.1**	C_18_H_32_O_3_	295.2278 (calcd. 295.2273)	−0.50	**277.2176**, 230.9864, 195.1398, 157.0871, 59.0144
**1.2**		295.2277 (calcd. 295.2273)	−0.82	277.2172, 211.1341, **183.1390**, 111.0814
**1.3**		295.2278 (calcd. 295.2273)	−0.63	277.2171, 195.1389, **183.1026**, 155.1077, 59.0140
**1.4**		295.2277 (calcd. 295.2273)	1.33	**277.2171**, 195.1394, 171.1025
**2**	C_18_H_30_O_3_	293.2120 (calcd. 293.2117)	−0.57	249.2221, 195.1387, 167.1074, **113.0971**, 59.0143
**3**	C_18_H_30_O_3_	293.2120 (calcd. 293.2117)	−0.09	249.2220, 195.1388, 179.1075, 167.1075, **113.0971**, 59.0141
**4**	C_18_H_30_O_3_	293.2120 (calcd. 293.2117)	−0.51	249.2222, 221.1542, 197.1176, **185.1181**, 125.0969
**5**	C_18_H_34_O_3_	297.2446 (calcd. 297.2430)	−0.15	**279.2327**, 155.1077, 141.1284, 59.0142
**6**	C_18_H_32_O_2_	279.2334 (calcd. 279.2324)	1.64	233.1544, 178.8147, **134.8652**, 96.9602, 83.0252, 68.9957
**7**	C_18_H_34_O_2_	281.2488 (calcd. 281.2481)	0.69	**136.8628**, 100.9336, 83.0251, 68.9965
Linoleic acid	C_18_H_32_O_2_	279.2329 (calcd. 279.2324)	1.64	233.1546, 178.8147, **134.8651**, 96.9600, 83.0254, 68.9965
Oleic acid	C_18_H_34_O_2_	281.2488 (calcd. 281.2481)	0.69	**136.8625**, 100.9335, 83.0254, 68.9957

^a^ Base peaks are indicated in bold font.

**Table 2 jof-08-01163-t002:** Fatty acid composition in four selected mushrooms (%).

Fatty Acids	*F. officinalis*	*E. tinctorium*	*A. flettii*	*O. tomentosa*
C11:1	0.0	0.0	0.2	0.0
C12:0	0.8	1.0	0.2	0.3
C12:1	0.5	0.5	0.0	0.1
C14:0	0.3	0.5	0.4	0.5
C14:1	0.0	0.6	0.1	0.2
C15:0	0.3	2.0	2.1	1.5
C15:1	0.7	0.7	0.0	0.2
C16:0	6.0	14.6	8.1	14.1
C16:1	0.3	2.1	0.9	0.7
C16:1-*trans*	0.3	0.9	0.3	0.2
C17:0	0.4	0.8	2.2	0.3
C17:1	0.0	1.1	0.0	0.0
C18:0	4.5	10.4	3.7	4.3
C18:1 (oleic)	57.1	34.7	22.8	23.6
C18:1 (vaccenic)	2.6	2.8	2.2	1.5
C18:2n6 (linoleic)	17.1	15.9	51.7	48.4
C18:2n6-*trans*	0.0	0.5	0.1	0.1
C18:3n3	0.0	0.1	0.2	0.2
C18:3n6	0.4	0.5	0.1	0.1
C18:4n3	0.4	0.3	0.0	0.0
C19:1	0.0	0.2	0.1	0.0
C20:0	0.3	0.9	0.4	0.3
C20:1n11-*cis*	0.4	0.6	0.2	0.3
C20:1n15-*cis*	3.4	1.2	0.2	0.2
C20:2n6	0.3	0.4	1.1	0.3
C20:3n6	1.0	0.9	0.1	0.2
C20:4n6	0.3	1.4	0.1	0.2
C20:5n3	0.0	0.3	0.1	0.3
C22:0	0.4	1.1	0.9	0.3
C22:1	0.3	0.2	0.3	0.1
C22:4n6	0.3	0.0	0.0	0.0
C22:5n6	0.6	0.0	0.0	0.0
C22:6n3	0.7	2.3	0.6	0.4
C24:0	0.2	0.5	0.1	0.1
C24:1	0.0	0.0	0.2	0.8
SFAs	13.2	31.9	18.1	21.7
UFAs	86.8	68.1	81.9	78.3
MUFAs	65.5	45.7	27.6	27.9
PUFAs	21.2	22.4	54.3	50.5
Omega-3 Fatty Acids	1.2	3.0	0.9	1.0
Omega-6 Fatty Acids	20.0	19.4	53.4	49.5
*trans* Fatty Acids	0.3	1.4	0.3	0.3

SFAs: ∑saturated fatty acids; UFAs: ∑unsaturated fatty acids; MUFAs: ∑monounsaturated fatty acids; PUFAs: ∑polyunsaturated fatty acids. Fatty acids at <0.2% of the total fatty acid yield were excluded.

## Data Availability

Not applicable.

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
