# Peer review of "Antiproliferative Fatty Acids Isolated from the Polypore Fungus Onnia tomentosa"

_jof, 2022, doi:10.3390/jof8111163_

Round 1

Reviewer 1 Report

- line 67: all other solvents and.... which are these solvents and reagents? and which is the reliable source?

-result 3.1: figure 1b is confused. the authors talk about inhibition action of E1, while the cell viability is increased through the days, reaching an increase of 50% in 7 days. please explain more

+ please explain more why did you choose the E1 and not some of the other E2 or E3? and more specific why did you choose the 0.5 mg/ml of E1, that appear the same inhibitory results with E2, E3 and not a bigger concentration where the difference is obvious?

- table 2 could be somehow not so chaotic if you remove the fatty acids that are not appeared or they appeared in extremely low concentrations in your analysis, like C6:0 etc. , as they don't play a role in the result. 

Author Response

Reviewer #1

Here are the comments from Reviewer 1 and our response to each of the comment.

Point 1: Line 67: all other solvents and….which are these solvents and reagents? And which is the reliable source?

Response 1: We thank Reviewer 1 for pointing this deficiency. We have now added more specific information on the solvents and reagents used and the companies that provided these items. This is on page 2 in the Materials and Chemicals section.

Point 2: Result 3.1: figure 1b is confused. The authors talk about inhibition action of E1, while the cell viability is increased through the days, reaching an increase of 50% in 7 days. Please explain more.

Response 2: We thank Reviewer 1 for questioning this result and its interpretation. We deliberately used an intermediate concentration of E1 (i.e., 0.5 mg/mL) to see if it would still be inhibitory over a 7-day period. As seen in Figure 1b, as compared to the control, the inhibitory effect can be seen starting at Day 2 and this inhibitory effect, as compared to the control, persisted for 7 days.  To address this issue, we have inserted “as compared to the control” to line 336 on page 5.

Point 3: Please explain more why did you choose the E1 and not some of the other E2 or E3? And more specific why did you choose the 0.5 mg/mL of E1, that appear the same inhibitory results with E2, E3 and not a bigger concentration where the difference is obvious?

Response 3: We thank Reviewer 1 for raising this point. We did not choose E2 and E3 because even at 1 mg/mL, the inhibitory effect was modest (only 20-25% inhibition).  We did not choose E4 (NaOH extract) because its inhibitory effect was only at 50% even at 1 mg/mL. In addition, we wish to isolate antiproliferative small molecules and E4 extract most likely contains only polysaccharides and not small molecules.  As mentioned in Response 2, we wish to see the effect of an intermediate concentration of E1 (i.e., 0.5 mg/mL) on proliferation of HeLa cells over 7 days as compared to the control. If we were to use 1 mg/mL of E1, we would expect to see at least 90% inhibition by day 1 or 2 as predicted by Reviewer 2. To address this issue, we have made the modification on page 5 lines 329-336 to better explain the rationale.

Point 4: Table 2 could be somehow not so chaotic if you remove the fatty acids that are not appeared or they appeared in extremely low concentrations in your analysis, like C6:0 etc., as they don’t play a role in the result.

Response 4: We thank Reviewer 1 for this excellent suggestion. As recommended, we have reduced the size of Table 2 by setting the criterion that fatty acids with less than or equal to 0.2% in any of the 4 mushrooms be removed. Similar changes were also made to Table S2.

Reviewer 2 Report

The manuscript Antiproliferative “Fatty Acids Isolated from the Polypore Fungus Onnia tomentosa” by Lee et al. looks as well-done research. However, there are some serious points that make it low valuable at this stage. 1) The data doesn’t reflect the title. For instance, the authors discuss the bioactivity of extracts and fatty acids content of four mushrooms. 2) google search on “fatty acid antitumor” revealed more than 1000000 references. It seems HeLa cells are not enough to make conclusions on antiproliferative activity of compounds. No positive control was presented. 3) google search on “autoxidation of linoleic acid” revealed 180000 results. Degradation of the fatty acids was studied very well. What is new? Just caution? I also would like pay attention that the simultaneous use of diethyl ether, dichloromethane and chloroform is no sense to make extraction of fungal metabolites because they have quite similar properties. Species identifications just using ITS sequencing is also inconsistent.

Author Response

Reviewer #2

Reviewer 2 commented that our manuscript looks as well-done research. However, Reviewer 2 also commented that there are serious points that make the manuscript of low value at this stage. Here are the comments from Reviewer 2 and our response to each of the comment.

Point 1: The data doesn’t reflect the title. For instance, the author discussed the bioactivity of extracts and fatty acids content of four mushrooms.

Response 1: We thank Reviewer 2 for raising this point. In this study, our antiproliferative-guided approach led to the isolation of linoleic acid, oleic acid and other linoleic oxygenated fatty acids from the fungus O. tomentosa. We showed some of the fatty acids to have antiproliferative activity.  Therefore, we feel that the existing title is appropriate.  Given that fatty acids are rich in O. tomentosa, we were interested to determine their overall content in the species as compared to other species. The latter study was an additional study and not the core investigation in our manuscript.

Point 2: Google search on “fatty acid antitumor” revealed more than 10000000 references. It seems HeLa cells are not good enough to make conclusions on antiproliferative activity of compounds. No positive control was presented.

Response 2: We thank Reviewer 2 for raising this point. Under the simple culture conditions (EMEM with 10% FCS) that we used for HeLa cells as described in this study and in several of our earlier studies using HeLa cells in comparison to other cancer cell lines [1-3], we have consistently observed similar pattern of antiproliferative activity exhibited by various compounds (small molecules and polysaccharides) on the different cancer cell lines. Based on our extensive experiences and consistencies in our various studies, we strongly believe that the fast-growing HeLa cells is an excellent and reliable model for studying potential antiproliferative compounds.

We have now included the MTT results of the positive control doxorubicin in Fig. S55 in the Supplementary Materials, and the comparison of its IC50 values with those of the fatty acids are described on page 13 lines 697-700.

Point 3: Google search on “autoxidation of linoleic acid” revealed 180000 results. Degradation of fatty acids was studied very well. What is new? Just caution?

Response 3: We thank Reviewer 2 for this critique. Yes, we agree that autoxidation of pure linoleic acid is a very well-studied topic. As described in the Discussion section, the autoxidation of fatty acids has not been described during their isolation from a natural source. Although this may seem trivia, such information is critical for researchers working in the field. In addition to caution researchers working in the field, we have added an implication of our findings on O. tomentosa; valuable information for future exploration as a possible edible and/or medicinal mushroom. This is described on page 16 line 881-884 in the Discussion and in line 24-26 in the Abstract.

Point 4: I also would like to pay attention that the simultaneous use of diethyl ether, dichloromethane and chloroform is no sense to make extraction of fungal metabolites because they have quite similar properties.

Response 4: We thank Reviewer 2 for this insight/critique. Extraction of the ethanolic extract (E1) of O. tomentosa was carried out using solvents of increasing polarity from non-polar hexane (polarity index: 0.0) to more polar solvents which are DEE (2.8), DCM (3.1), and CHCl3 (4.1), ensuring the extraction of a wide range of compounds with different polarity. Among the four organic solvents used, HEX (6.9%) gave the highest yield and was later found to contain fatty acids as the major component. Unexpectedly, HPLC analysis on DEE, DCM, and CHCl3 showed that they contained the same fatty acids, indicating that ethanolic extract from O. tomentosa was rich in fatty acids. Hence, subsequent small molecule isolation was focused only on HEX and DEE extracts.

These results were very different from another study performed in our lab on the fungus Echinodontium tinctorium [4]. Sequential phase extraction of the methanolic extract of E. tinctorium with HEX gave the lowest yield (1.6%) as compared to CHCl3 (20%) and EA (37.5%) extracts. Even though CHCl3 and EA extracts have close polarity indexes of 4.1 and 4.4, respectively, they showed different HPLC chromatographic profiles. Subsequent purification of EA extracts resulted in one phenol derivative (MW: 124 Da) and one diphenylmethane derivative (MW: 260 Da). CHCl3 extract detected different set of small molecules with m/z 271 [M+H]+, m/z 439 [M+H]+, and m/z 543 [M+H]+. Thus, based on our experiences, using different solvents with quite similar properties can help in extracting different types of small molecules.

Point 5: Species identifications just using ITS sequencing is also inconsistent.

Response 5: We thank Reviewer 2 for pointing this out. Based on our experiences and research so far on using genetic analysis in identifying mushroom species, determining the DNA sequences at the ITS region is an acceptable and successful approach in identifying mushroom species [1-3,5,6]. To our knowledge, this is the standard and acceptable approach in the field.

References

[1] Yaqoob A, et al. (2020) Grifolin, neogrifolin and confluentin from the terricolous polypore Albetrellus flettii suppress KRAS expression in human colon cancer cells. PLOS ONE 15(5): e0231948.

[2] Barad A, et al. (2018) Anti-proliferative activity of a purified polysaccharide isolated from the basidiomycete fungus Paxillus involutus. Carbohydrate Polymers 181: 923-930.

[3] Zeb M, et al. (2022) Isolation and characterization of anti-proliferative polysaccharide from the North American fungus Echinodontium tinctorium. Scientific Reports 12: 17298.

[4] Zeb M. Bioactive polysaccharides and small molecules from the native north American fungus Echinodontium Tinctorium [Dissertation]. University of Northern British Columbia, Prince George, BC; 2021: 145

[5] Smith A, et al. (2017) Growth-inhibitory and immunomodulatory activities of wild mushrooms from north-central British Columbia (Canada). International Journal of Medicinal Mushrooms 19(6): 485-497.

[6] Deo G, et al. (2019) Antiproliferattive, immunostimulatory, and anti-inflammatory activities of extracts derived from mushrooms collected in Haida Gwaii, British Columbia, Canada. International Journal of Medicinal Mushrooms 21(7): 629-643.

Reviewer 3 Report

The manuscript entitled (Antiproliferative Fatty Acids Isolated from the Polypore Fungus Onnia tomentosa) by Lee et al. discussed the isolation, purification, and structure elucidation of of eight known linoleic oxygenated fatty acids (1.1-1.4, 2-5) together with linoleic and oleic acids by HPLC-MS/MS and 1D/2D NMR analyses of from the EtOH extract of O. tomentosa led to the identification. Antiproliferative and immunomodulatory activities have been evaluated.

The manuscript is good, and it could be accepted after covering the following issues

1- English-editing is needed. There are many typing and grammatical mistakes. Remove the pronouns ``we, our, …..`` and rephrase the sentences.

2- In Abstract

please add short sentences about the biological results. Add the fungus family also in keywords

3- In material and chemicals

HeLa human cervical cancer should write human cervical cancer cell (HeLa)

4- Are these fungi edible, what about toxicity of them?

5- Please clarify and compare the biological data of compounds with the positive control used. What is the value of each control i.e. example doxorubicin???

6- The MS sections should be arranged according to the journal Guidelines.

7- Ensure that all references have been cited in the MS.

8- Authors should highlight the significance of the reported compounds and their expected impact of them.

Author Response

Reviewer #3

Reviewer 3 commented that our manuscript is good and it could be accepted after covering the issues raised. Following are the comments by Reviewer 3 and our responses to each of the comment.

Point 1: English-editing is needed. There are many typing and grammatical mistakes. Remove the pronouns “we, our, …” and rephrase the sentence.

Response 1: We thank Reviewer 3 for pointing this out and the suggestion. We have removed the pronouns and rephrase the sentences as recommended by Reviewer 3. We have also used an editorial service to check on the written English. When we merged the track changes from the editing service with our track changes document, the manuscript looked very complicated and messy.  Therefore, we have manually incorporated the editorial changes from the editing service. 

Point 2: In Abstract, please add short sentences about the biological results. Add the fungus family also in keywords.

Response 2: We thank Reviewer 3 for this suggestion. We have added the fungus family into the keywords and short sentence in the Abstract to summarize the biological data.

Point 3: In Materials and Chemicals, HeLa human cervical cancer should write human cervical cancer cell (HeLa).

Response 3: We have made the correction as suggested by Reviewer 3

Point 4: Are these fungi edible, what about toxicity of them?

Response 4: We thank Reviewer 3 for this question. Onnia tomentosa has never been reported to be consumed by humans, at least in the literature, and we have no idea about its toxicity.

Point 5: Please clarify and compare the biological data of compounds with the positive control used. What is the value of each control i.e. example doxorubicin??

Response 5: We thank Reviewer 3 for pointing this out.  We have now included the MTT results from the positive control doxorubicin (Fig. S55) in the Supplementary Materials. We have also included a sentence on page 13 lines 697-700 to compare the IC50 values of the fatty acids with the positive control doxorubicin.

Point 6: The manuscript section should be arranged according to the journal guidelines.

Response 6: We have checked and ensured that the manuscript is arranged as according to the journal guidelines.

Point 7: Ensure that all references have been cited in the manuscript.

Response 7: We thank Reviewer 3 for this suggestion. We have checked that all references are cited in the manuscript.

Point 8: Authors should highlight the significance of the reported compounds and their expected impact of them.

Response 8: We thank Reviewer 3 for this important suggestion. Results of the fatty acids contents and their bioactivity reported in our study have implication on future exploration of O. tomentosa as a possible edible and/or medicinal mushroom. To address this, we have inserted two sentences on page 16 line 881-884 in the Discussion section and line 24-26 in the Abstract.

Round 2

Reviewer 2 Report

Point 1: The data doesn’t reflect the title. For instance, the author discussed the bioactivity of extracts and fatty acids content of four mushrooms.

Response 1: We thank Reviewer 2 for raising this point. In this study, our antiproliferative-guided approach led to the isolation of linoleic acid, oleic acid and other linoleic oxygenated fatty acids from the fungus O. tomentosa. We showed some of the fatty acids to have antiproliferative activity.  Therefore, we feel that the existing title is appropriate.  Given that fatty acids are rich in O. tomentosa, we were interested to determine their overall content in the species as compared to other species. The latter study was an additional study and not the core investigation in our manuscript.

Reviewer opinion 1. The source of known fatty acids does not matter if you study the bioactivity. If you wanted to compare fatty acids content in different fungi it is better to use relative species.

Point 2: Google search on “fatty acid antitumor” revealed more than 10000000 references. It seems HeLa cells are not good enough to make conclusions on antiproliferative activity of compounds. No positive control was presented.

Response 2: We thank Reviewer 2 for raising this point. Under the simple culture conditions (EMEM with 10% FCS) that we used for HeLa cells as described in this study and in several of our earlier studies using HeLa cells in comparison to other cancer cell lines [1-3], we have consistently observed similar pattern of antiproliferative activity exhibited by various compounds (small molecules and polysaccharides) on the different cancer cell lines. Based on our extensive experiences and consistencies in our various studies, we strongly believe that the fast-growing HeLa cells is an excellent and reliable model for studying potential antiproliferative compounds.

We have now included the MTT results of the positive control doxorubicin in Fig. S55 in the Supplementary Materials, and the comparison of its IC50 values with those of the fatty acids are described on page 13 lines 697-700.

 Reviewer opinion 2. The data are not original and novel. For obtaining new data you should evaluate new bioactivities of fatty acids or use a broader number of cells lines.

Point 3: Google search on “autoxidation of linoleic acid” revealed 180000 results. Degradation of fatty acids was studied very well. What is new? Just caution?

Response 3: We thank Reviewer 2 for this critique. Yes, we agree that autoxidation of pure linoleic acid is a very well-studied topic. As described in the Discussion section, the autoxidation of fatty acids has not been described during their isolation from a natural source. Although this may seem trivia, such information is critical for researchers working in the field. In addition to caution researchers working in the field, we have added an implication of our findings on O. tomentosa; valuable information for future exploration as a possible edible and/or medicinal mushroom. This is described on page 16 line 881-884 in the Discussion and in line 24-26 in the Abstract.

 Reviewer opinion 3. The data may be useful but did not relate to low antiproliferative activity. What does indicate fatty content and its low activity: is the fungus potentially edible or medical?  

Point 4: I also would like to pay attention that the simultaneous use of diethyl ether, dichloromethane and chloroform is no sense to make extraction of fungal metabolites because they have quite similar properties.

Response 4: We thank Reviewer 2 for this insight/critique. Extraction of the ethanolic extract (E1) of O. tomentosa was carried out using solvents of increasing polarity from non-polar hexane (polarity index: 0.0) to more polar solvents which are DEE (2.8), DCM (3.1), and CHCl3 (4.1), ensuring the extraction of a wide range of compounds with different polarity. Among the four organic solvents used, HEX (6.9%) gave the highest yield and was later found to contain fatty acids as the major component. Unexpectedly, HPLC analysis on DEE, DCM, and CHClshowed that they contained the same fatty acids, indicating that ethanolic extract from O. tomentosa was rich in fatty acids. Hence, subsequent small molecule isolation was focused only on HEX and DEE extracts.

These results were very different from another study performed in our lab on the fungus Echinodontium tinctorium [4]. Sequential phase extraction of the methanolic extract of E. tinctorium with HEX gave the lowest yield (1.6%) as compared to CHCl3 (20%) and EA (37.5%) extracts. Even though CHCl3 and EA extracts have close polarity indexes of 4.1 and 4.4, respectively, they showed different HPLC chromatographic profiles. Subsequent purification of EA extracts resulted in one phenol derivative (MW: 124 Da) and one diphenylmethane derivative (MW: 260 Da). CHCl3 extract detected different set of small molecules with m/z 271 [M+H]+m/z 439 [M+H]+, and m/z 543 [M+H]+. Thus, based on our experiences, using different solvents with quite similar properties can help in extracting different types of small molecules.

Reviewer opinion 4. I think that DCM data may be deleted.  

Point 5: Species identifications just using ITS sequencing is also inconsistent.

Response 5: We thank Reviewer 2 for pointing this out. Based on our experiences and research so far on using genetic analysis in identifying mushroom species, determining the DNA sequences at the ITS region is an acceptable and successful approach in identifying mushroom species [1-3,5,6]. To our knowledge, this is the standard and acceptable approach in the field.

Reviewer opinion 5. The use of DNA sequences of the ITS region for species identification alone is a very bad practice leading to many false results that should be eliminated from mycological papers.

Author Response

We are grateful to Reviewer 2 for his/her insightful comments and critiques for improving our manuscript. We have carefully addressed all the concerns raised by Reviewer 2 in the revised manuscript. Our text modifications to the original manuscript are specifically described in the pages below, and the changes were marked up using the “Track Changes” function.  Following are point-by-point responses to Reviewer 2’s comments/critiques.

Reviewer #2

Reviewer 2 commented that our manuscript looks as well-done research. However, Reviewer 2 also commented that there are serious points that make the manuscript of low value at this stage. Here are the comments from Reviewer 2 and our response to each of the comment.

Point 1: The data doesn’t reflect the title. For instance, the author discussed the bioactivity of extracts and fatty acids content of four mushrooms.

Response 1: We thank Reviewer 2 for raising this point. In this study, our antiproliferative-guided approach led to the isolation of linoleic acid, oleic acid and other linoleic oxygenated fatty acids from the fungus O. tomentosa. We showed some of the fatty acids to have antiproliferative activity.  Therefore, we feel that the existing title is appropriate.  Given that fatty acids are rich in O. tomentosa, we were interested to determine their overall content in the species as compared to other species. The latter study was an additional study and not the core investigation in our manuscript.

Reviewer opinion 1: The source of known fatty acids does not matter if you study the bioactivity. If you wanted to compare fatty acids content in different fungi it is better to use relative species.

Response to Reviewer opinion 1: We thank Reviewer 2 for pointing this out.  To take Reviewer 2’s suggestion and to address this issue, we have included the fatty acids contents of related Inonotus species (I. hispidus, I. obliquus and I. radiatus) in Table S3. The text describing these are on page 16 line 625-631.

Point 2: Google search on “fatty acid antitumor” revealed more than 10000000 references. It seems HeLa cells are not good enough to make conclusions on antiproliferative activity of compounds. No positive control was presented.

Response 2: We thank Reviewer 2 for raising this point. Under the simple culture conditions (EMEM with 10% FCS) that we used for HeLa cells as described in this study and in several of our earlier studies using HeLa cells in comparison to other cancer cell lines [1-3], we have consistently observed similar pattern of antiproliferative activity exhibited by various compounds (small molecules and polysaccharides) on the different cancer cell lines. Based on our extensive experiences and consistencies in our various studies, we strongly believe that the fast-growing HeLa cells is an excellent and reliable model for studying potential antiproliferative compounds.

We have now included the MTT results of the positive control doxorubicin in Fig. S55 in the Supplementary Materials, and the comparison of its IC50 values with those of the fatty acids are described on page 13 lines 697-700.

Reviewer opinion 2: The data are not original and novel. For obtaining new data you should evaluate new bioactivities of fatty acids or use a broader number of cell lines.

Response to Reviewer opinion 2: We agree with Reviewer 2 that the findings of linoleic acid and oleic acid possessing antiproliferative activity are not novel.  The novelty of our study is the discovery of these and other related fatty acids being the major antiproliferative chemical constituents in the mushroom species Onnia tomentosa using bioactivity-guided approach. We have highlighted this novelty in the Discussion on page 14 line 518-522, in the Conclusion section on page 16 line 646-649, and in the Abstract on page 1 line 22-23. Another novelty in this study is the first description of fatty acid contents in four mushroom species. To highlight this, we have modified the sentence on page 12 line 468-469.

The major focus of our study is on finding antiproliferative compounds in O. tomentosa.  Finding new bioactivities for fatty acids is not the focus of our study. 

We agree that assessing linoleic acid and oleic acid for antiproliferative activity in a broader number of cell lines is worthy of investigations based on our findings. However, such studies had already been done. In the Discussion from page 15 line 579 to page 16 line 610, we have described these published studies.

Point 3: Google search on “autoxidation of linoleic acid” revealed 180000 results. Degradation of fatty acids was studied very well. What is new? Just caution?

Response 3: We thank Reviewer 2 for this critique. Yes, we agree that autoxidation of pure linoleic acid is a very well-studied topic. As described in the Discussion section, the autoxidation of fatty acids has not been described during their isolation from a natural source. Although this may seem trivia, such information is critical for researchers working in the field. In addition to caution researchers working in the field, we have added an implication of our findings on O. tomentosa; valuable information for future exploration as a possible edible and/or medicinal mushroom. This is described on page 16 line 881-884 in the Discussion and in line 24-26 in the Abstract.

Reviewer opinion 3: The data may be useful but did not relate to low antiproliferative activity. What does indicate fatty acid content and its low activity: is the fungus potentially edible or medical?

Response to Reviewer opinion 3: Onnia tomentosa has never been explored as an edible or medicinal mushroom. Therefore, our findings on its fatty acid contents and their antiproliferative are useful information for possible future exploration in this area of study. The text on page 16 line 631-634 addressed this issue.

Point 4: I also would like to pay attention that the simultaneous use of diethyl ether, dichloromethane and chloroform is no sense to make extraction of fungal metabolites because they have quite similar properties.

Response 4: We thank Reviewer 2 for this insight/critique. Extraction of the ethanolic extract (E1) of O. tomentosa was carried out using solvents of increasing polarity from non-polar hexane (polarity index: 0.0) to more polar solvents which are DEE (2.8), DCM (3.1), and CHCl3 (4.1), ensuring the extraction of a wide range of compounds with different polarity. Among the four organic solvents used, HEX (6.9%) gave the highest yield and was later found to contain fatty acids as the major component. Unexpectedly, HPLC analysis on DEE, DCM, and CHCl3 showed that they contained the same fatty acids, indicating that ethanolic extract from O. tomentosa was rich in fatty acids. Hence, subsequent small molecule isolation was focused only on HEX and DEE extracts.

These results were very different from another study performed in our lab on the fungus Echinodontium tinctorium [4]. Sequential phase extraction of the methanolic extract of E. tinctorium with HEX gave the lowest yield (1.6%) as compared to CHCl3 (20%) and EA (37.5%) extracts. Even though CHCl3 and EA extracts have close polarity indexes of 4.1 and 4.4, respectively, they showed different HPLC chromatographic profiles. Subsequent purification of EA extracts resulted in one phenol derivative (MW: 124 Da) and one diphenylmethane derivative (MW: 260 Da). CHCl3 extract detected different set of small molecules with m/z 271 [M+H]+, m/z 439 [M+H]+, and m/z 543 [M+H]+. Thus, based on our experiences, using different solvents with quite similar properties can help in extracting different types of small molecules.

Reviewer opinion 4: I think that DCM data may be deleted.

Response to Reviewer opinion 4: We thank Reviewer 2’s suggestion.  We have now deleted all the DCM data in the Supplementary Materials and the text describing the data.

Point 5: Species identifications just using ITS sequencing is also inconsistent.

Response 5: We thank Reviewer 2 for pointing this out. Based on our experiences and research so far on using genetic analysis in identifying mushroom species, determining the DNA sequences at the ITS region is an acceptable and successful approach in identifying mushroom species [1-3,5,6]. To our knowledge, this is the standard and acceptable approach in the field.

Reviewer opinion 5: The use of DNA sequences of the ITS region for species identification alone is a very bad practice leading to many false results that should be eliminated from mycological papers.

Response to Reviewer opinion 5: We agree with Reviewer 2 that fungal species identification using ITS region alone could lead to false results. This is especially the case if the DNA sequences obtained are not sufficed to discriminate between species [1] or there is not sufficient resolution for species identification [2], as discussed in detail in the two references [1,2] on fungal identification.  However, this is not the case for the four mushroom species described in our study where around 99% similarities were found. Both references had indicated that ITS region works well in most cases and remains the universal fungal barcode marker. Both references also indicated that when there is not sufficient resolution for species identification, then additional methods will be required.

In addition to achieving high resolution (~99% similarity) species identification using the ITS region, we have also used morphological characteristics to confirm the species identity. This is mentioned in section 2.2 on page 2 line 84-85.

References

[1] Lucking, R. et al. (2020) Unambiguous identification of fungi: where do we stand and how accurate and precise is fungal DNA barcoding? IMA Fungus 11:14.

[2] Raja, HA. et al. (2017) Fungal identification using molecular tools: a primer for the natural products research community. J Nat Prod 80(3):756-770.

Reviewer 3 Report

No comments

Author Response

Reviewer 3 has no further comments.